# Gelatin/Hyaluronic Acid Content in Hydrogels Obtained through Blue Light-Induced Gelation Affects Hydrogel Properties and Adipose Stem Cell Behaviors

**DOI:** 10.3390/biom9080342

**Published:** 2019-08-05

**Authors:** Shinji Sakai, Hiromi Ohi, Masahito Taya

**Affiliations:** Department of Materials Science and Engineering, Graduate School of Engineering Science, Osaka University, 1-3 Machikaneyama, Toyonaka, Osaka 560-8531, Japan

**Keywords:** gelatin, hyaluronic acid, tissue engineering, hydrogels, bioprinting, stem cells, biomedical, blue light, hydrogelation

## Abstract

Composite hydrogels of hyaluronic acid and gelatin attract great attention in biomedical fields. In particular, the composite hydrogels obtained through processes that are mild for cells are useful in tissue engineering. In this study, hyaluronic acid/gelatin composite hydrogels obtained through a blue light-induced gelation that is mild for mammalian cells were studied for the effect of the content of each polymer in the precursor solution on gelation, properties of resultant hydrogels, and behaviors of human adipose stem cells laden in the hydrogels. Control of the content enabled gelation in less than 20 s, and also enabled hydrogels to be obtained with 0.5–1.2 kPa Young’s modulus. Human adipose stem cells were more elongated in hydrogels with a higher rather than lower content of hyaluronic acid. Stem cell marker genes, *Nanog*, *Oct4*, and *Sox2*, were expressed more in the cells in the composite hydrogels with a higher content of hyaluronic acid compared with those in the hydrogel composed of gelatin alone and on tissue culture dishes. These results are useful for designing conditions for using gelatin/hyaluronic acid composite hydrogels obtained through blue light-induced gelation suitable for tissue engineering applications.

## 1. Introduction

Biopolymers are polymers obtained from natural resources such as microorganisms and plants, and they are used in a wide range of applications. Biocompatible biopolymers are excellent candidates to be used in pharmaceutical and medical applications including tissue engineering and regenerative medicine [1,2,3,4]. Gelatin and hyaluronic acid (HA) are biopolymers that have been historically prized for their usefulness in clinical fields [5,6,7]. In the tissue engineering field, their effectiveness for supporting growth and elevating biological functions of mammalian cells are well known [8,9,10]. Therefore, a wide variety of scaffolds have been prepared from not only gelatin or hyaluronic acid alone but also from multiple polymers, including both gelatin and hyaluronic acid [11,12,13]. The ability to mimic the architecture of the extracellular matrix has been reported for gelatin/hyaluronic acid hybrid hydrogels [13,14]. Scaffolds consisting of multiple polymers are attractive because these scaffolds can have multiple functions, which cannot be obtained from single polymer scaffolds [15,16,17]. The degree of expression of the functions attributed to each polymer is controllable by altering the ratio of the polymers. In addition, altering the ratio of polymers can cause a change in the mechanical properties of three-dimensional (3D) microenvironment surrounding the cells. Recently, it has been well recognized that cell behaviors, including the differentiation of stem cells, can be influenced by the mechanical properties of the 3D microenvironment [18,19]. Scaffolds consisting of both gelatin and hyaluronic acid also have been prepared for culturing varieties of cells such as vascular endothelial cells [20], myoblasts [21], osteoblasts [22], and stem cells [8,23,24,25]. 

Various methods have been applied for obtaining insoluble forms of these polymers from polymer solutions. One method is that of cross-linking the polymers in aqueous solution through visible light irradiation in the presence of photoredox catalysts [26,27,28]. Photo-induced cross-linking is attractive because the progress of the reaction is easily controllable by on/off switching of photoirradiation. In addition, using visible light is more suitable than using ultraviolet light for obtaining hydrogels that contain cells. The exposure of cells to ultraviolet light risks inducing chromosomal and genetic instability [29]. Various visible light-mediated hydrogelation systems have been reported for obtaining cell-laden hydrogels [30,31]: Mazaki et al. prepared bone marrow-derived stromal cell-laden hydrogels by irradiating visible light to cell-suspending furfurylamine-conjugated gelatin solution containing rose bengal [30]. Fenn et al. prepared mesenchymal stem cell-laden hydrogels by irradiating visible light to cell-suspending methacrylate hyaluronic acid solution containing eosin Y and triethanolamine [32]. Ruthenium(II) tris-bipyridyl dication ([Ru(bpy)_3_]^2+^) is a photoredox catalyst that catalyzes the cross-linking of phenolic hydroxyl moieties, including those in tyrosine residues under blue light irradiation (Figure 1) [28,33,34,35,36]. This system has been used for obtaining hydrogels containing fibroblasts [37] and carcinoma cells without applying bioprinting [35]. 

Recently, we reported the potency of the blue light-mediated cross-linking system for fabricating cell-laden hyaluronic acid/gelatin hybrid hydrogels [28]. The hybrid hydrogels were obtained from derivatives of hyaluronic acid and gelatin, both possessing phenolic hydroxyl (Ph) moieties (gelatin-Ph and HA-Ph). Ph moieties were cross-linked through the blue light-mediated reaction. The aim of this study was to investigate the effect of the content of gelatin-Ph and HA-Ph in precursor solutions on gelation behavior, properties of hydrogels, and stem cell marker gene expression of human adipose-derived stem cells (hASCs) enclosed in the hydrogels (Figure 1). It is recognized well that hydrogels are bioactive materials that regulate stem cell fate [38]. Detailed studies on the effect of the content of gelatin-Ph and HA-Ph have not been performed in previous bioprinting work. In this study, we investigated the effects under the condition of fixing one polymer concentration and changing the other polymer concentration based on the condition of the previous study for bioprinting (3.0% gelatin-Ph/0.5% HA-Ph) [28]. The knowledge obtained in this study would further enhance the utility of hybrid hydrogels obtained through blue light-mediated cross-linking in a variety of tissue engineering applications. ASCs are multipotent stem cells and are widely used in research because they can be obtained in abundance with minimal invasiveness [39]. It has been reported that hyaluronic acid and gelatin support the maintenance of differentiation potentials [40] and proliferation [41] of ASCs, respectively. 

## 2. Materials and Methods

### 2.1. Materials

Gelatin (bovine skin, ca. 225 g bloom, type B) and sodium hyaluronic acid (HA, average molecular weight, approximately 1000 kDa) were purchased from Kewpie (Tokyo, Japan) and Sigma-Aldrich (St. Louis, MO, USA), respectively. Tyramine hydrochloride and Tris(2,2′-bipyridyl)dichlororuthenium(II) hexahydrate [Ru(bpy)_3_]Cl_2_·6H_2_O were obtained from Tokyo Chemical Industry (Tokyo, Japan). Sodium ammonium persulfate (SPS) and *N*-hydroxy sulfosuccinimide were purchased from Wako Pure Chemical Industries (Osaka, Japan). 1-Ethyl-3-(3-dimethylaminopropyl) carbodiimide (EDC) was obtained from the Peptide Institute (Osaka, Japan). Gelatin-Ph (2.3 × 10^−4^ mol-Ph/g) and HA-Ph (1.2 × 10^−4^ mol-Ph/g) were synthesized through conjugation with 3-(4-hydroxyphenyl)propionic acid and tyramine hydrochloride, respectively, using EDC and *N*-hydroxy sulfosuccinimide according to previous reports [42,43]. The contents of introduced Ph moieties were determined based on the absorbance at 275 nm (Appendix A) using a calibration curve obtained by measuring the absorbance of known percentages of tyramine hydrochloride. The polymeric aqueous solutions used in the study were obtained by dissolving Gelatin-Ph and HA-Ph at 1.0%–5.0% and 0%–0.75%, respectively, in phosphate-buffered saline (PBS, pH 7.4). The contents of [Ru(bpy)_3_]^2+^ and SPS in the solutions were fixed at 1.0 and 2.0 mM. hASCs were obtained from Lonza (Walkersville, MD, USA) and cultured in growth medium for ASCs (PT-4505, Lonza) containing 10% (v/v) fetal bovine serum. 

### 2.2. Viscosity and Gelation Time of Solutions

The viscosities of the solutions and the gelation times were measured using a rheometer (HAAKE MARS III, Thermo Fisher Scientific, Waltham, MA, USA) equipped with a parallel plate geometry. A top plate with a 25 mm radius was set at a 0.5 mm height from a transparent glass bottom plate, and solution (0.7 mL) was put between the plates. The transitions of storage elastic moduli and loss elastic moduli with time were measured with irradiating blue light (0.45 W/m^2^ at a 452 nm wavelength) through the glass bottom at 25 °C. Blue LED light (Microscope LED Blue Ring Light, Srate, Nanyang, China) was used as the light source. Gelation time was defined as the time it took to achieve the intersection point of the transitions of the two moduli [44]. Measurements were performed in triplicate for each condition.

For evaluating the effect of the gelation speed on printability, the spreading of the solutions extruded from a 27 gauge stainless-steel needle onto a glass stage was determined by measuring the width of the gelled filaments. The solution was extruded at 22 mm/s at the tip of the needle onto a stage moving at 22 mm/s under blue light irradiation (0.45 W/m^2^ at 452 nm). Measurements were performed on eight specimens for each condition. 

### 2.3. Mechanical Properties and Molecular Permeabilities of Hydrogels

Hydrogels with a 12 mm diameter and 3.5 mm thickness were obtained by irradiating 0.4 mL of solution, poured into the molds, with blue light (0.45 W/m^2^ at 452 nm) for 20 min. The irradiation time was decided based on the result of a preliminary study showing that 15 min was necessary to achieve plateau storage modulus (*G*’) at conditions giving 35 s of gelation time (Appendix A). The hydrogels were compressed using a table-top material tester (EZ-Test, Shimadzu, Kyoto, Japan) at 1.0 mm/min. Young’s modulus was calculated from the data for less than 10% compression. Measurements were performed on four specimens for each condition. 

Diffusion coefficients of 4, 10, and 70 kDa fluorescein isothiocyanate (FITC)-labeled dextran in hydrogels were determined as reported previously [45]. In brief, hydrogels were soaked in PBS containing FITC-dextran for 24 h at room temperature. The diffusion coefficients of FITC-labeled dextrans were determined by the method of fluorescence recovery after photobleaching (FRAP) based on the measurement using confocal laser scanning microscopy (CLSM, Nikon C2 Plus, Tokyo, Japan). Measurements were performed five times for each condition.

### 2.4. Human Adipose-Derived Stem Cell (hASC) Studies

hASCs were suspended in polymer solutions at 5.0 × 10^5^ cells/mL. The solutions were poured into a 24-well tissue culture plate at 0.4 mL/well after covering the bottom of each well with 1.0% agarose gel to prevent cell adhesion. Then, blue light (0.45 W/m^2^ at 452 nm) was irradiated for 20 min. After several rinses with medium, the specimens enclosing the hASCs were incubated in a medium. At 1, 7, and 14 d of culture, cells were stained with Calcein-AM and propidium iodide (PI) to evaluate the viability. The morphology of the cells at a 400 to 500 μm height from the bottom of the specimens was observed using CLSM after staining the cells with Calcein-AM. To monitor the expression of the stemness-related genes—*Nanog*, *Oct4*, and *Sox2*—in hASCs, RT-PCR studies were performed at 1, 7, and 14 d of culture for the cells collected from the hydrogels by degrading using the solution containing hyaluronidase (1000 U/mL) and trypsin (0.025 *w/v*%). Total RNAs of the collected hASCs and adhered cells in cell culture dishes were extracted with RNeasy Plus Micro Kits (Qiagen, Hilden, Germany) according to the manufacturer’s instructions. The total RNA concentration was determined by optical density at 260 nm using a spectrophotometer (e-spect, BM Equipment, Tokyo, Japan). cDNA was generated from the extracted RNA using a SuperScript™ VILO™ cDNA Synthesis Kit (Invitrogen, Carlsbad, CA, USA). The target sequences from the resultant cDNA templates were then amplified and quantified using a qPCR system (Promega, Madison, WI, USA) and previously reported primers [46]. Briefly, qPCR was performed in triplicate for each condition. The expression level was analyzed and normalized to β-actin. The relative quantity of gene expression was calculated with the cells obtained before enclosing them in hydrogels as the reference. 

### 2.5. Statistical Analysis

The data shown represent the mean and standard deviation. Comparisons between two datasets were made using an unpaired *t*-test. Values of *p* < 0.05 were considered significant. 

## 3. Results and Discussion

### 3.1. Solution Viscosities

First, we studied the effect of the content of gelatin-Ph and HA-Ph on the viscosity–shear rate profiles of solutions. As shown in Figure 2a, the viscosity of solutions at a low shear rate (between 0.01 and 1 s^−1^) increased with increasing HA-Ph content at a fixed gelatin-Ph content (3.0%). The viscosities of solutions containing 0.5% and 0.75% HA-Ph were around 1000 and 5500 mPa s at a shear rate of 0.01 s^−1^. At higher shear rate values, viscosities of these solutions decreased and showed similar values at around 50^−1^ (Figure 2b), indicating an enhancement of the shear-thinning property by increasing the HA-Ph content. The contribution of hyaluronic acid on the enhancement of the shear-thinning property has been reported in previous studies [47,48,49]. The shear-thinning property, it is argued, occurs when the rate of breakdown of the stiff intermolecular network junctions exceeds the rate of reformation of associations and entanglements [47]. 

At a fixed content of HA-Ph (0.5%), as shown in Figure 2, the viscosity of solutions at a low shear rate (between 0.01 and 1 s^−1^) decreased when the content of gelatin-Ph increased from 1.0% to 3.0%. The trend was reversed when the gelatin-Ph content further increased to 5.0%. The viscosity of 5.0% gelatin-Ph solution at 0.01 s^−1^ was about three times larger than that of 3.0% gelatin-Ph solution. The result of the reversal of the trend in the transition of viscosity with increasing gelatin-Ph content is attributed to the change in the degree of the intermolecular interactions between the polysaccharide and protein [50,51,52]. A bimodal curve was established, which separates the region of co-solubility from the region of separation for the effect of the concentrations of the protein and polysaccharide [52]. The above result would be helpful for predicting the cellular damage caused by shear stress during the handling of precursor solutions and the force required when the precursor solution flows in capillaries and needles [53]. In addition, the result relating to the shear-thinning property is useful for fabricating 3D constructs with good printability in an extrusion-based 3D printing [54]. 

### 3.2. Gelation Behavior

The time necessary for gelation of precursor solutions under the exposure of blue light (450 W/m^2^ at 452 nm) decreased with increasing gelatin-Ph content and HA-Ph content at fixed HA-Ph (0.5%) and gelatin-Ph (3.0%), respectively (Figure 3) (*p* < 0.05). The shortest gelation time was 6.8 ± 0.8 s at 3.0% gelatin-Ph + 0.75% HA-Ph. The decrease in gelation time with an increase in the total amount of polymer-Ph resulted from the increase of the content of Ph moieties in the reaction system. Typically, the enzyme reaction rate increases with increasing substrate concentration, except in the range of excess substrate concentration. Similar results of a decrease in gelation time with increasing the content of Ph moieties in solutions have been reported for solutions containing a variety of single polymer-Ph [42,55,56]. The conditions that provide a gelation time of less than 10 s are useful for in situ fabrication of hydrogels in vivo by injecting precursor solutions [57,58] with irradiation of blue light. The concentrations of [Ru(bpy)_3_]^2+^ and SPS also govern the gelation time [28]. In this study, their concentrations were fixed (1.0 mM [Ru(bpy)_3_]^2+^ and 2.0 mM SPS) based on a previous result, indicating a successful enclosure of mammalian cells without decreasing their viability [28]. 

In 3D bioprinting applications, the time necessary for gelation and the viscosity of solution govern the printability. We evaluated the effect on 3D printing by determining the stabilization of solutions extruded from a needle with a 220 μm inner diameter and 400 μm outer diameter with an exposure of blue light (0.45 W/m^2^ at 452 nm). As shown in Figure 4, the width of the gelled filament extruded from the needle at the same linear velocity as the movement speed of the stage decreased with increasing the total content of gelatin-Ph and HA-Ph (Figure 4). A value far from 1.0 means the solution spread before gelation. The trend of decreasing hydrogel filament width with increasing the content of gelatin-Ph and HA-Ph was consistent with the trend obtained for viscosity (Figure 2) and gelation time (Figure 3): hydrogel filaments with smaller widths were obtained at the conditions giving a higher viscosity at a low shear rate and shorter gelation time. The smallest relative width of hydrogel filaments to the needle diameter was 1.02 ± 0.04 at 3.0% gelatin-Ph + 0.75% HA-Ph. The value was 1.05 ± 0.04 at 3.0% gelatin-Ph + 0.5% HA-Ph. The reason the hydrogel filament width converged to the outer diameter of the needle was due the extruded solution contacting the tip wall of the needle before being placed on the stage [59]. 

### 3.3. Hydrogel Properties

The mechanical properties of hydrogels play vital roles in their mechanical stability toward external forces as well as their behaviors when contacting cells, including the differentiation of stem cells [60,61]. As shown in Figure 5, stiffer hydrogels formed from the solution with higher polymer-Ph contents, except for the trend when increasing the HA-Ph content from 0.5% to 0.75% at 3.0% gelatin-Ph (*p* = 0.40). The smallest and the largest Young’s moduli were 0.26 ± 0.05 at 1.0% gelatin-Ph + 0.5% HA-Ph and 1.14 ± 0.10 kPa at 5.0% gelatin-Ph + 0.5% HA-Ph, respectively. The polymer content-dependent increase of Young’s modulus is explained by the increase in the density of cross-linked polymers. The results for the diffusion coefficients of 4, 10, and 70 kDa FITC-labeled dextran in the hydrogels (Figure 6) well support this explanation: smaller diffusion coefficients were detected for the hydrogels with a higher Young’s modulus. A polymer network offers transport resistance for solutes, and a denser network offers higher transport resistance [62]. 

### 3.4. Human Adipose Stem Cell Behaviors

Subsequently, we studied the effect of the content of gelatin-Ph and HA-Ph on the viability, morphology, and stem cell marker gene expression of hASCs enclosed in gelatin-Ph/HA-Ph hydrogels. Figure 7a shows the transition of the viability of hASCs during 14 d of culture in hydrogels. All viabilities at 1 d of culture were almost identical, more than 95%, for all the conditions. Except for the condition of 5.0% gelatin-Ph + 0.5% HA-Ph, the viabilities were more than 90%, even at 7 d of culture. These results demonstrate that blue light-induced gelation did not give a significant adverse effect that would induce cell death on the enclosed hASCs independent of the hydrogel composition. A notable result was that the viabilities of the cells in 5.0% gelatin-Ph + 0.5% HA-Ph hydrogel at 7 and 14 d of culture were clearly lower than those in other hydrogels (*p* < 0.05). This result is attributed to the higher mechanical stress given to the cells by the surrounding hydrogels and the lower permeability in the hydrogel than those in other hydrogels (Figure 3 and Figure 4). A similar result of a decrease in cell number in stiffer hydrogels was reported for human embryonic stem cells enclosed in hydrogels composed of HA-Ph alone [63]. Yang et al. indicated an increase in hASC apoptosis in a hydrogel composed of gelatin alone, with an increase in the polymer concentration resulting in an increase in hydrogel stiffness [41]. They demonstrated that the enclosed hASCs in such hydrogels increased in the first 10–14 d of culture but then decreased [41]. 

The confocal images of hASCs in hydrogels at 7 d of culture demonstrate suppression of cellular growth in 1.0% gelatin-Ph + 0.5% HA-Ph and 2.0% gelatin-Ph + 0.5% HA-Ph hydrogels compared to other hydrogels (Figure 7b). This result means a certain amount of gelatin is necessary for improving the spreading of enclosed hASCs. A notable result was the difference in cell morphology in hydrogels. The hASCs in 3.0% gelatin-Ph + 0.5% HA-Ph were the most elongated (Figure 7b: 3.0–0.5). In contrast, the hASCs in 3.0% gelatin-Ph hydrogels (Figure 7b: 3.0–0), without HA-Ph, were the least elongated. The less elongated morphology of enclosed hASCs is consistent with hASCs enclosed in hydrogels composed of gelatin alone obtained through transglutaminase-mediated gelation [41]. The difference in the cellular morphology clearly demonstrates the function of HA-Ph for elongation of the hASCs. Hyaluronic acid binds to its cluster determinant 44 (CD44) receptor [8,20,64]. In addition, the binding of hyaluronic acid and CD44 enhances adhesion, migration, and cell–cell interaction in varieties of cells, including hASCs [8,20]. The greater elongation of the cells in 3.0% gelatin-Ph + 0.5% HA-Ph hydrogel (Figure 7b: 3.0–0.5) than those in 3.0% gelatin-Ph + 0.75% HA-Ph hydrogel (Figure 7b: 3.0–0.75), despite the similar stiffness and molecular permeability of the hydrogels (Figure 5 and Figure 6), also indicates the composition-dependent elongation of the enclosed hASCs. Microenvironmental changes caused by proteolytic degradation of gelatin-Ph by hASC-secreted metalloprotease may have induced the difference. The degradation of matrix surrounding cells provides the space for cell elongation and proliferation [38,65]. Lei et al. reported that mouse mesenchymal stem cells can only spread in Arg-Gly-Asp (RGD)-modified HA hydrogel with proteolytic degradable character than in nondegradable hydrogel [66]. Due to the lower content of HA-Ph, 3.0% gelatin-Ph + 0.5% HA-Ph hydrogel would have provided more space for cell elongation. Investigating the microenvironmental changes during cell culture would give important information for understanding the results. 

Figure 8 shows the transition of the expression of stem cell marker genes—*Nanog*, *Oct4*, and *Sox2*—in the hASCs during 14 d of culture in the hydrogels. It has been reported that the expression of *Nanog*, *Oct4*, and *Sox2* genes are an indicator of stemness of hASCs [67]. Except for the cells in the hydrogel composed of gelatin-Ph alone (Figure 8b), all cells showed higher expressions of the stem cell marker genes compared to those cultured in the tissue culture dish (relative expression > 1, *p* < 0.05). The degrees of the expression were higher in the cells containing a smaller amount of gelatin-Ph at 0.5% HA-Ph (Figure 8a) and a higher amount of HA-Ph at 0.3% gelatin-Ph (Figure 8b). In addition, the degrees increased with increasing the period of culture. These results demonstrate that HA-Ph cross-linking through blue light-induced gelation is effective for the preservation of stem cell properties in hASCs in the same way as with native hyaluronic acid [25,64]. Regarding the mechanism of the upregulation of the stem cell marker gene expression by hyaluronic acid, it has been explained as a result of the enhancement of the expression of CD44, a receptor of hyaluronic acid [64]. This explanation supports the result of the upregulation of stem cell marker gene expression with increasing HA-Ph content. Regarding the decrease in stem cell marker gene expression with increasing gelatin-Ph, it can be explained by the increase in the degree of the intermolecular interaction between HA-Ph with gelatin-Ph, which caused the decrease in viscosity with increasing gelatin-Ph from 1.0% to 3.0% in 0.5% HA-Ph solution (Figure 2). The intermolecular interaction would inhibit or lower the interaction between HA-Ph and CD44. The differentiation of enclosed ASCs was not examined in this study, but the capacity of differentiation was reported in a previous study for hASCs enclosed in 3.0% gelatin-Ph + 0.5% HA-Ph hydrogel [28]. The effects of the content of gelatin-Ph and HA-Ph on differentiation to specific cell types, such as adipocytes, chondrocytes, myocytes, osteoblasts, and neurocytes, are currently under investigation.

## 4. Conclusions

In this study, we examined hyaluronic acid/gelatin composite hydrogels obtained through blue light irradiation in the presence of [Ru(bpy)_3_]^2+^ and SPS from the viewpoint of their applications in tissue engineering. In detail, we investigated the effects of the content of gelatin-Ph and HA-Ph in PBS on the viscosity and gelation behavior of solutions, including the printability, mechanical and molecular permeability properties of resultant hydrogels, and hASC behaviors in the hydrogels. The contents of gelatin-Ph and HA-Ph greatly influenced their behaviors, that is, these features can be controlled by altering concentrations of these polymers without changing the intensity of blue light and the concentration of [Ru(bpy)_3_]^2+^ and SPS. An increase in the content of polymers did not always result in a solution with higher viscosity. It was necessary to determine the content, taking into account the influence of the interaction between the polymers for obtaining the solution with an intended viscosity. The time necessary for gelation changed in the range around 5–20 s. The Young’s modulus of hydrogels changed in the range of 0.2–1.2 kPa. The hASCs in the hydrogels containing HA-Ph became more elongated than those in the hydrogel obtained from gelatin-Ph alone (Figure 7a). In addition, the degrees of stem cell marker gene expression of *Nanog*, *Oct4*, and *Sox2*, were upregulated in the composite hydrogels compared to those in the hydrogel containing gelatin-Ph alone and the tissue culture dish (Figure 8). Especially, the degree of upregulation was enhanced with an increase in the HA-Ph content. Recently, the influence of the mechanical signal on cell behaviors, such as proliferation, elongation, and differentiation, has become widely recognized [19]. In the fabrication processes of cell-laden hydrogels, the viscosity and gelation rate of precursor solutions govern the damage to cells [53] and the ease in handling. This means that the composition of precursor solutions has to be decided based on a comprehensive view. We believe the results obtained in this study will be helpful in future works using blue light-mediated gelation in tissue engineering applications.

## Figures and Tables

**Figure 1 biomolecules-09-00342-f001:**
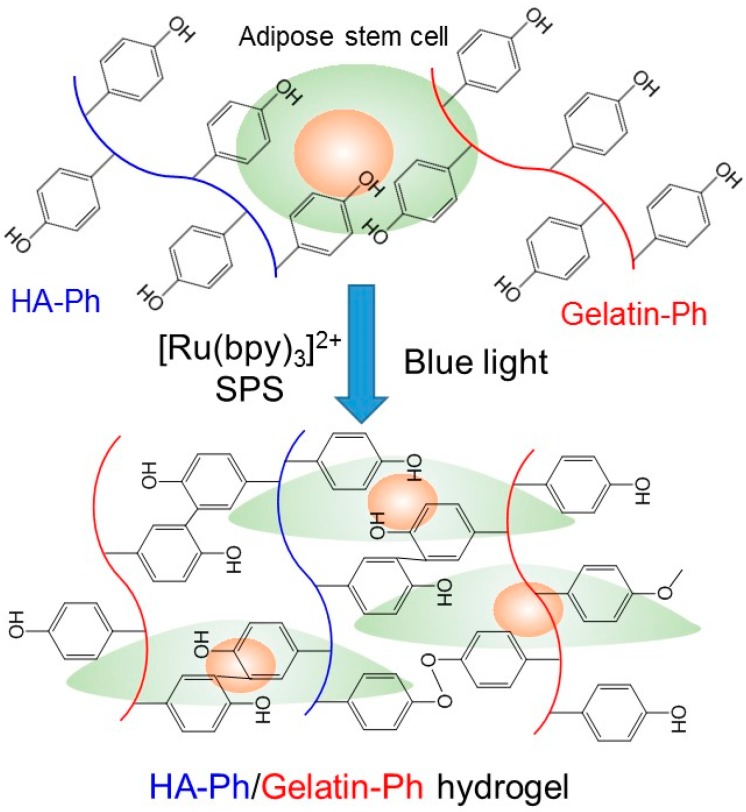
Schematic representation of hyaluronic acid (HA)-Ph/gelatin-Ph hydrogel formation via blue light irradiation in the presence of [Ru(bpy)_3_]^2+^ and sodium ammonium persulfate (SPS).

**Figure 2 biomolecules-09-00342-f002:**
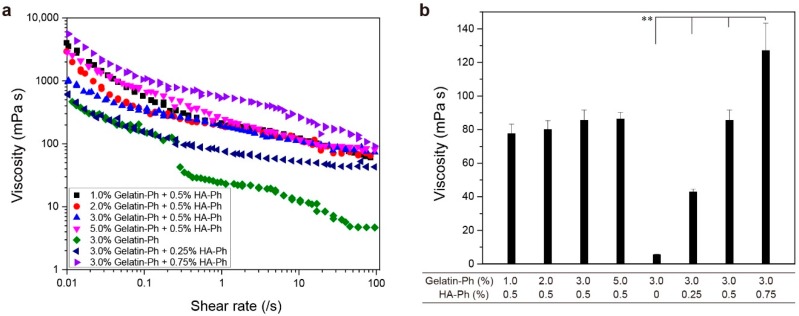
(**a**) Effects of gelatin-Ph and HA-Ph content on the viscosity profiles of solutions as a function of the shear rate. (**b**) The viscosity of solutions at a 45 s^−1^ shear rate. Bars: S.D. (*n* = 4). ** *p* < 0.01.

**Figure 3 biomolecules-09-00342-f003:**
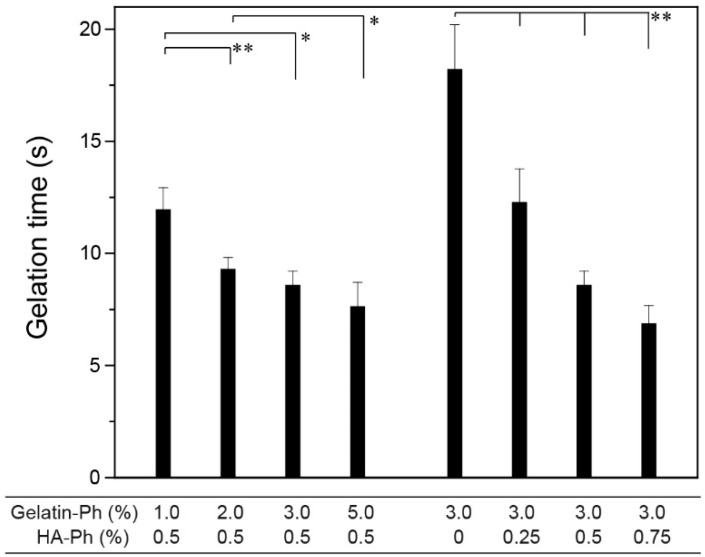
Effects of gelatin-Ph and HA-Ph content on gelation time. Bars: S.D. (*n* = 4). * *p* < 0.05, ** *p* < 0.01.

**Figure 4 biomolecules-09-00342-f004:**
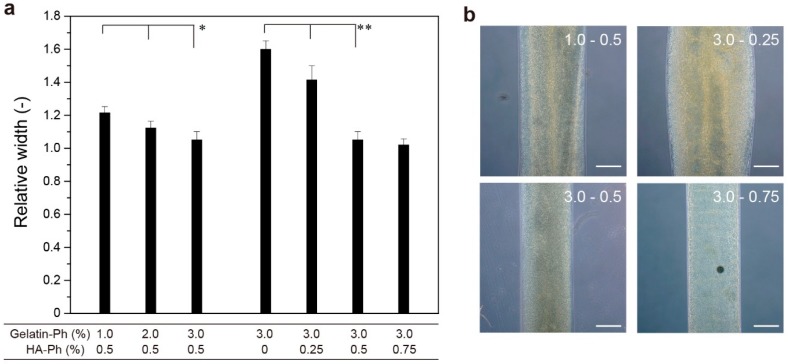
Effects of gelatin-Ph and HA-Ph content on the width of hydrogel filaments extruded on a stage. (**a**) Values normalized by an outer diameter of an extrusion needle (400 μm). The linear velocity of the solution at the tip of the needle and the movement speed of the stage were set to be the same. Bars: S.D. (*n* = 8). * *p* < 0.05, ** *p* < 0.01. (**b**) Microphotographs of hydrogel filaments at typical conditions. Bars: 200 μm.

**Figure 5 biomolecules-09-00342-f005:**
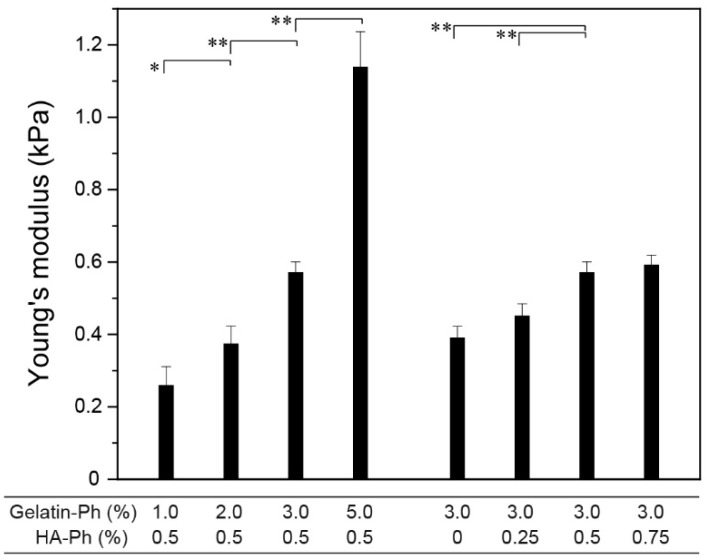
Effects of gelatin-Ph and HA-Ph content on Young’s modulus of hydrogels. Bars: S.D. (*n* = 4). * *p* < 0.05, ** *p* < 0.01.

**Figure 6 biomolecules-09-00342-f006:**
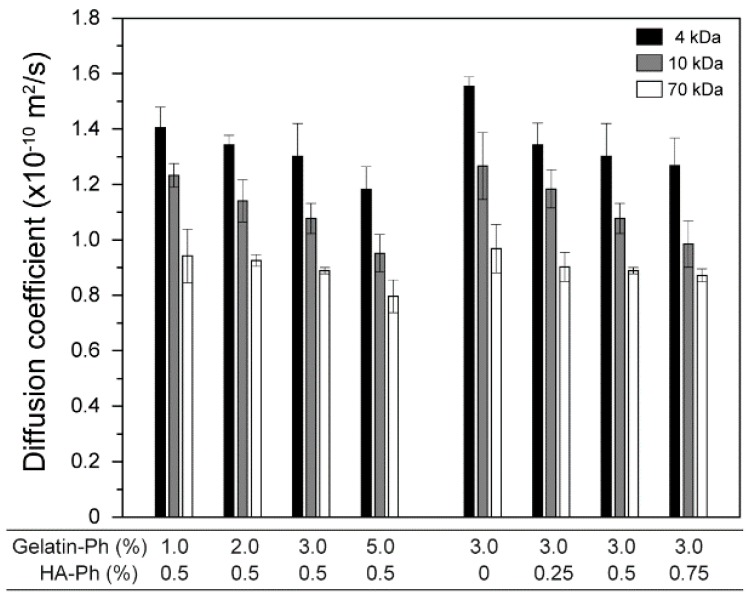
Effects of gelatin-Ph and HA-Ph content on diffusion coefficients of 4, 10, and 70 kDa fluorescein isothiocyanate (FITC)-labeled dextrans in hydrogels. Bars: S.D. (*n* = 4).

**Figure 7 biomolecules-09-00342-f007:**
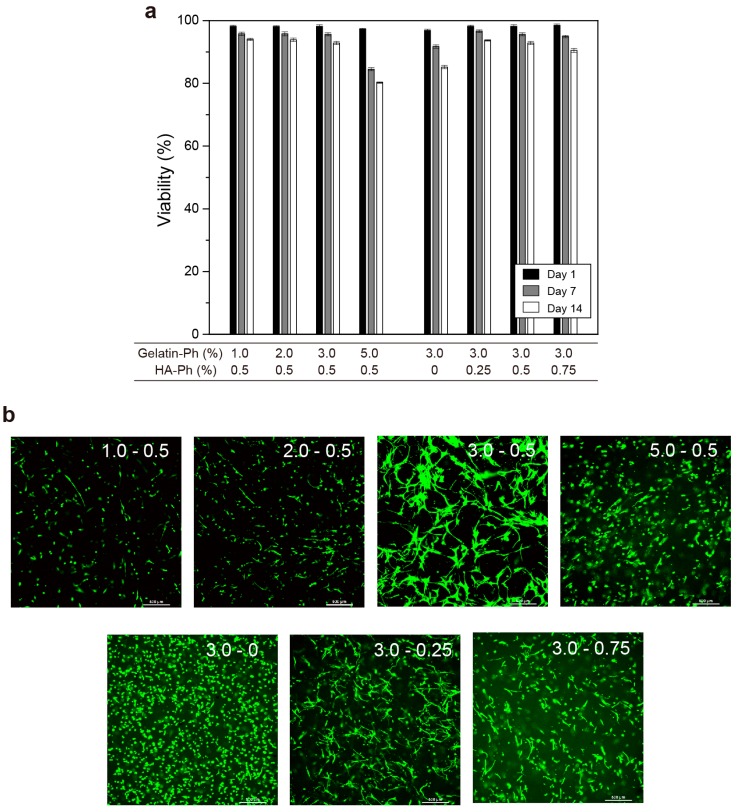
Effects of gelatin-Ph and HA-Ph content on (**a**) viability of hASCs during 14 days of culture (Bars: S.D., *n* = 3) and (**b**) morphology of enclosed hASCs at 7 days of culture (Bars: 500 μm). Confocal fluorescence images in panel b were taken at a 400 to 500 μm height from the bottom of the specimens (1 mm thickness) at 7 days of culture. The numerical characters in each image, X and Y, mean the contents of gelatin-Ph (%) and HA-Ph (%), respectively.

**Figure 8 biomolecules-09-00342-f008:**
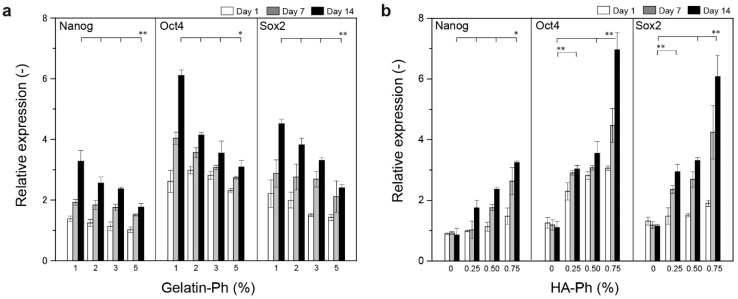
Effects of gelatin-Ph and HA-Ph content on the expression of stem cell marker genes, *Nanog*, *Oct4*, and *Sox2*. Cell-laden hydrogels were prepared by varying (**a**) gelatin-Ph content at 0.5% HA-Ph, and (**b**) HA-Ph content at 3.0% gelatin-Ph. Bars: S.D. (*n* = 3). The relative expressions were calculated with the cells obtained before enclosing hydrogels as the reference. * *p* < 0.05, ** *p* < 0.01.

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
