# Peer review of "Gelatin/Hyaluronic Acid Content in Hydrogels Obtained through Blue Light-Induced Gelation Affects Hydrogel Properties and Adipose Stem Cell Behaviors"

_biomolecules, 2019, doi:10.3390/biom9080342_

Round 1

Reviewer 1 Report

In this manuscript, authors have investigated the effect of polymer content in composite precursor solutions on their gelatin behaviors under blue light-induced gelation followed by their cellular properties. . This study is of a great interest and may be considered for possible publication in Biomolecules journal. Manuscript is well written and characterized. However, I gone through this research manuscript and found some critical points to be considered before acceptance of this manuscript, as follows:

1.      Introduction section regarding the use of any polymer or biopolymer is superficial and should be elaborated for the required background for the researchers or research professionals. In addition, authors should incorporate recent articles based on gelatin and/or hyaluronic acid for various potential applications in tissue engineering along with challenges for dynamic three-dimensional microenvironment for tissue regeneration. Authors may include following recent articles for the background of this study as Carbohydrate polymers180, 2018, 128-144; Chemical Engineering Journal317, 2017, 119-131; Carbohydrate polymers193, 2018, 228-238; Composites Science and Technology175, 2019, 35-45, etc.

2.      The variation in the degree of crosslinking in gelatin/hyaluronic acid composite hydrogels should have been calculated. In addition, structural characterization (NMR or FTIR, XRD) may also be included in support of these concluding results.

3.      Authors should include the digital images (if available) of the extruded polymer-gels, while printing. It will provide better information for further research perspectives. In addition, actual mechanical plots of molded hydrogels should be incorporated in the manuscript, including all mechanical data in a Table form.

In my opinion, this manuscript can be accepted after this major revision.

Reviewer 2 Report

The manuscript reports on the preparation and characterization of 3D cell-laden hyaluronic acid and gelatin hydrogels via blue light photopolymerization, which is of interest for tissue engineering applications, allowing surpassing common limitations associated to the use of UV light. The experiments were in general well-conducted, but the presentation and discussion of the data should be improved, along with some key specific issues. Please see questions/comments bellow.

Major issues:

1. Considering that the Ru/SPS system is already used for visible light photocrosslinking and the gelatin/HA material was previously reported for bioprinting hydrogels loaded with human adipose stem cells using an optimal hydrogel formulation (0.5% HA‐Ph and 3.0% Gelatin‐Ph,  Biopolymers. 2018, 109, 2, e23080), the originality and novelty of the work must be clarified. In addition, the rational for selecting the gelatin and HA concentrations should be explained in the text.

2. The introduction is too general missing in providing enough details regarding the topic of the work. For instance, current hydrogel systems that allow photocrosslinking under the visible light are missing along with key aspects that regulate stem cell fate within 3D hydrogels. Authors should consider reworking these aspects.

3. Statistical analysis is missing in the manuscript and must be included.

4. In section 3.2, the effect of gelation kinetics on the printability of hydrogel precursors is evaluated and correlated with the viscosity. However, throughout the manuscript there is no further testing or demonstration of the printing using the hydrogels. In the context of the manuscript and considering that the printing of HA/gelatin hydrogels was previously reported, why this specific test was performed? Still related to bioprinting, did the hydrogel precursors form a stable filament at the nozzle? Considering that the minimum gelation time is 6.8 ± 0.8 s and the printing speed leads to a faster polymer deposition, hydrogel formulations should form a stable filament at the nozzle to prevent spreading onto the substrate during the deposition.

5. On the abstract, authors state that hydrogels can be formed in less than 20 s with the shortest gelation time to be 6.8 ± 0.8 s (page 6, line 186). However, in Materials and Methods (section 2.3) hydrogels were prepared with 20 min of irradiation. Authors should clarify these issues and explain why 20 min was used for hydrogel formation. Gelation kinetic graphs could also be included in supplementary material to provide more details regarding the required time for the G’ and G’’ to achieve the plateau.

6. The stem cell fate within engineered 3D hydrogel networks is affected by several microenvironmental factors including the network degradability, stiffness and biochemical composition. In their experimental setting, authors prepared several formulations with different mechanical properties, variable biochemical composition and degradability, to evaluate how gelatin/HA content affect stem cell spreading and the maintenance of stem cell markers. Considering that gelatin inherently contains cell adhesion/cell-degradable motifs and control HA hydrogels (w/o gelatin) are missing, small variations in the gelatin content will change not only the hydrogel stiffness, but also the biochemical composition, making difficult to state broad conclusions from these experiments without appropriate controls.

    - Regarding cell viability and spreading studies (Fig. 7), results show that 3% gelatin/0.5% HA promoted an elongated cell morphology, while either removing HA or increasing the HA content to 0.75% inhibits cell elongation. Why 0.75% HA in gelatin hydrogels, but not 0.5%, inhibits cell elongation despite their similar stiffness (Fig. 5)? These distinct cell responses must be further discussed in the manuscript and compared to other works using 3D hydrogels.

    - Comparing the hydrogels 3% gelatin/ 0.5% HA and 3% gelatin w/o HA, it is clear that cell spreading in higher when HA is present, though the higher elastic modulus. Did the authors expect that after 7d of culture hASCs displayed an almost round morphology within gelatin hydrogels w/o HA of relatively low stiffness? A control hydrogel made of gelatin w/o HA, but with similar stiffness to the 3%/0.5% hydrogel should be included to determine whether the observed effects are exclusively due to the presence of HA or can also arise from the hydrogel stiffness.

    - Gene expression studies show that higher HA content led to enhanced stem cell marker expression, while the opposite was verified for gelatin. Considering that higher HA content (0.75%) inhibits cell elongation, but favors stem cell marker expression, how did the authors explain these effects? To confirm that HA is the major player in the maintenance of stemness, a control experiment should be performed using 3% gelatin/ 0.75% HA hydrogels but blocking the CD44 receptor. This will elucidate whether the expression of stem cell markers is reduced in the presence of HA. As control gelatin hydrogels display lower stiffness, it is difficult to determine whether distinct cell responses are also a result of the gel stiffness. Another useful control is using HA hydrogel w/o gelatin, but with a similar stiffness to 3% gelatin/ 0.75% HA hydrogel in order to elucidate whether gelatin is necessary to maintain/enhance stem cell marker expression in the HA hydrogel.

Minor issues

1. In figure 2, the representation of y-axis in logarithmic scale will probably provide a more clear data interpretation. Also in these data, why the viscosity of HA alone was not included in the graph similarly to gelatin? To facilitate comparison, authors could eventually provide a Table with the viscosity of the tested solutions at a selected shear rate.

2. In the mechanical properties, why HA hydrogels alone were not tested similarly to gelatin at 3% w/o HA? The viscous modulus and/or phase angle could also be provided in the graphs to elucidate the contribution of HA to the viscoelastic behaviour of hydrogels. 

3. The following details should be included in Materials and Methods:

    - The degree of phenolic hydroxyl group substitution of hyaluronic acid and gelatin as well as the MW of HA;

    - More details on the LED source and the protocol used for cell recovery, namely the hyaluronidase and trypsin concentrations as well as the digestion conditions.

4. Check the sentence in page 7, line 213: “gelation time (Figure 3): A hydrogel”

5. In page 11, line 282 authors state that “a certain amount of gelatin is necessary for improving the proliferation of enclosed hASCs”. However, results only show cell viability and morphology. How cell proliferation was determined? This must be checked.

Round 2

Reviewer 1 Report

Authors have responded well against the Reviewer's comments and improved manuscript accordingly. In my opinion, this manuscript now can be accepted in this journal.

Author Response

Thank you for your valuable comments. Your comments in Round 1 were helpful for improving our manuscript. 

Reviewer 2 Report

Comment 1: This is the reviewer reply to a previous comment.

5. In page 11, line 282 authors state that “a certain amount of gelatin is necessary for improving the proliferation of enclosed hASCs”. However, results only show cell viability and morphology. How cell proliferation was determined? This must be checked.

Reply. In this study, cell-laden hydrogels were prepared at the same cell density. Therefore, we describe that a certain amount of gelatin is necessary for improving cellular proliferation based on Figure 7b.

Reviewer: Figure 7b is a calcein-AM and propidium iodide staining providing information regarding the ratio between live and dead cells, but not regarding cell proliferation. Although cellular hydrogels were prepared at the same cell density, hydrogels with different composition and elastic modulus often result in variable cell loss to the culture medium due to the network relaxation effect during the swelling. There are several reliable ways to determine cell proliferation within hydrogels such as counting the retrieved cells or using specific assays like ki67 staining and BrdU assay. These methods provide quantitative data regarding cell proliferation, allowing to establish reliable comparisons between time points and/or hydrogels. I suggest the authors to perform some quantitative assay to determine cell proliferation using the tested hydrogel systems. Alternatively, I would say that: a certain amount of gelatin is necessary for improving the spreading of enclosed hASCs.

Comment 2: carefully check the spelling and grammar in the manuscript. E.g., line 329: “Microenvirmantal change caused”.
